# Constrained Reinforcement Learning using Bender's Decomposition and Exact Constraint Satisfaction

## Abstract

Recent advancements in reinforcement learning (RL) have expanded its applications beyond sequential decision-making to encompass non-sequential tasks, such as matrix decompositions, automatic generation of sorting networks, and combinatorial optimization. However, these tasks often require problem-specific algorithm designs to ensure the validity of the solution. To address this limitation, we propose a universal framework that reformulates non-sequential tasks as constrained RL problems by learning to generate cutting planes, i.e., mathematical constraints that systematically refine the solution space. We ensure constraint satisfaction throughout the training process, enabling safe and efficient training even during deployment. We show the efficacy of our framework on two complex optimization problems: a reward-maximizing stochastic job-shop scheduling problem and a nonlinear, nonconvex packing problem. Our method achieves near-globally optimal solutions while accelerating convergence by up to a factor of 800.

## 1 Introduction

Reinforcement Learning (RL) is a powerful technique for solving challenging Markov Decision Processes (MDPs) through interaction with an environment. Historically, RL has been used for traditional control and robotics problems (see e.g., oh Kang et al. (2023)). However, as of recent, more effort has been put into using RL techniques to solve problems that do not naively fit into the MDP framework: One particularly famous example of this is AlphaTensor (Fawzi et al., 2022), which finds good factorizations of high dimensional tensors by reducing the problem to a "Tensor Game", where in each "turn" of the game three vectors are drawn which are used to build up a rank-1 tensor decomposition. A follow-up work "AlphaTensor Quantum" (Ruiz et al., 2024), extended the Tensor Game by introducing additional rules that reward decompositions which can be implemented easily on quantum computers. AlphaDev (Mankowitz et al., 2023) uses a similar synthetic game construction to discover new sorting algorithms by iteratively building up sorting networks (Knuth, 1998) which can be automatically verified.

In general, all of these approaches can be viewed as trying to optimize a reward $\mathcal{R}(s)$ over a set of constraints $s \in \mathcal{C}$. For instance, AlphaTensor's optimization problem can be concisely described as

$$\min T \tag{1a}$$

$$A = \sum_{i=0}^{T} a_i \otimes b_i \otimes c_i \tag{1b}$$

$$T \in \mathbb{N}, a_i, b_i, c_i \in \mathbb{R}^{N \times M \times K}, \tag{1c}$$

where $A$ is the tensor we want to decompose, $a_i, b_i, c_i$ are the coefficients of the rank-1 tensor decomposition, and $T$ is the number of rank-1 terms needed. While exact, this problem is not practically solvable: This problem is equivalent to a large combinatorial optimization problem over the natural numbers. AlphaTensor tries to find a minimum-rank decomposition by reducing this problem to an iterative minimization of the residual $\left\| A - \sum_{i=0}^{T} a_i \otimes b_i \otimes c_i \right\|$ and selecting $a_i, b_i, c_i$ using Monte-Carlo Tree-Search (MCTS). Reward is assigned once the model has found a point satisfying the constraints proportional to Eq. 1a.

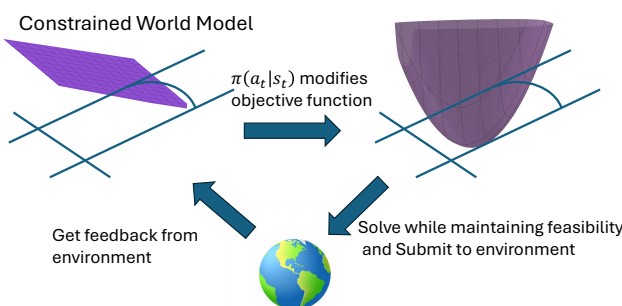

Figure 1: Overview of our method. We assume the existence of a (potentially NP-hard) world model over which we can optimize. Our policy $\pi$ modifies the objective function to give the desired behavior. The solution is given to the environment, which provides feedback (such as a stochastic events) to the world model, which can then be re-solved with the added information

Although this is a good way to model this specific problem, it is not a general way to solve constrained design problems. One core issue with sequential decompositions is that they can easily lead to a dead-end. While this cannot happen for the AlphaTensor problem (since $a_i, b_i, c_i$ are unconstrained), this does occur in SAT or Constraint Satisfaction Problems (CSPs). In fact, this is what makes e.g. SAT or CSP NP-complete: finding a point which satisfies the constraints is already hard. Assuming $P \neq NP$, no static policy without search will be able to solve an arbitrary instance of these types of problems. Further, due to the design of AlphaTensor and similar MCTS-based approaches they can only handle a *finite number of discrete actions* $a_i, b_i, c_i$.

Classically, Constrained RL (CRL) Problems have been modeled using the framework of Constrained Markov Decision Processes (CMDP) (Altman, 1999), which introduce a cost function $c(x)$ whose expected cumulative (discounted) sum has to be below a threshold $C$:

$$\max_{\pi} \sum_{i=0}^{T} \gamma^i R(x_i) P(x_{i+1}|x_i, a_i) \pi(a_i|x_i) \quad \text{s.t.} \quad \mathbb{E}\left[\sum_{i=0}^{T} \gamma^i c_i(x_i)\right] \leq C_i \quad (2a)$$

This modeling has the same disadvantages as the sequential decomposition of AlphaTensor. If finding a single point $x$ that satisfies the constraints is already NP-complete, there does not exist a static policy that solves this problem. Such hard constraints appear frequently in real world problems.

Further, *the CMDP framework does not allow for the modeling of hard constraints*: For example, if we have $C = 25$ and our method produces 20 trajectories with cost 30, and 10 trajectories with cost 0, the policy would be considered safe despite violating the cost upper bound twice as often as it stays feasible! In this work, we study *exact feasibility* where the RL agent has to maintain the constraint *every time*, not just in expectation. This allows us to frame design problems - such as AlphaTensor - as CRL problems, which is impossible using the CMDP framework. In general, the "safety in expectation" formulation used in CMDPs is incompatible with high risk or high complexity scenarios.

This work covers the scenario where hard constraint satisfaction for every instance has to be upheld during inference and *training*. For this, we assume access to the constraint set $\mathcal{C}$. This is a reasonable assumption in design problems such as AlphaTensor, Process Planning, Job-Scheduling, and similar other problems where conditions for feasible instances are known, but optimality is unknown. Knowledge of constraints is also mandatory if one wants to guarantee that a constraint is never violated during training: As is noted by (Müller et al., 2024), if the constraints are unknown one has to experience an action violates the constraint at least once before the agent can try to avoid it!

With techniques from mathematical programming we parameterize our policy *relative* to the feasible set $\mathcal{C}$, and hence guarantee the agent stays within the feasible region during inference *and* training, which allows for *fully online training with exact safety guarantees*. We do this by proposing a universal parameterization using an implicitly defined policy that iteratively improves its own objective function using cutting planes inspired by the framework of Bender's decomposition (see

Fig. 1). By moving the complexity of strict feasibility preservation into a dedicated solver such as SCIP (Bestuzheva et al., 2021) our method can scale to complex instances without sacrificing the expressivity of our neural network. We showcase our method by applying it to a highly constrained nonconvex nonlinear discrete packing problem (Sec. 5.1), and a complex reward-maximizing job scheduling problem (Sec. 5.2) including stochastic effects in the reward function.

## 2 RELATED WORK

From the point of view of constrained reinforcement learning, prior work mostly considers the setting of feasibility budgets where the constraint violations have to stay below a certain threshold (see Eq. 2). Constrained Policy Optimization (CPO) (Achiam et al., 2017) maintains constraints by descending inside the intersection of a trust region and the feasible set, using recovery steps when the policy is outside the feasible set. This method does not have a guarantee to be safe during training and cannot natively handle "no-regret" constraints. Another sometimes competitive approach is penalizing constraint violations with large negative values inside the reward, such as Fixed Penalty Optimization (FPO) (Achiam et al., 2017). Tessler et al. (2018) uses a more sophisticated version of FPO by dynamically adjusting the penalty parameter $\lambda$ during optimization. However, neither of these models handle hard "no-regret" constraints or guarantee feasibility during training. Dalal et al. (2018) consider continuous action spaces where safety is guaranteed by projecting onto a feasible set of safe actions. While they consider hard constraints, they can only operate in continuous convex action spaces. Similar to our method, they also delegate their safety constraint to a Quadratic Programming (QP) solver to compute the projection onto the feasible set. Our method has the advantage of not being limited to QP-solvers and being able to deal with combinatorial settings.

To our knowledge, the only other work considering the absolute "no-regret" setting is Müller et al. (2024), which showcase an algorithm that minimizes the per-episode error rather than the expected error. Unfortunately, their method does not scale beyond small state and action spaces (in their case 5 states and 5 actions). Further, their method cannot guarantee safety during training, as one needs to try every action in every state to absolutely guarantee the safety under an unknown constraint function. We extend their ideas of "no regret learning" by considering the case where constraints are known, allowing us to be safe even during training.

From the point of view of combinatorial optimization and RL, Bello et al. (2016) consider solving the traveling salesman (TSP) and Knapsack problem using specialized neural networks. However, they do not consider learning from a stochastic or nonlinear environments, and assume variable selection cannot lead to infeasible dead-ends. For instance, Bello et al. (2016) uses the fact that their TSP instances live on a fully connected graph, meaning that one can arbitrarily pick any order of nodes and will still get a possible tour. If one had a sparsely connected graph, Bello et al. (2016)'s method no longer works as picking certain node orders can get the agent into a dead-end.[1]

## 3 BACKGROUND: BENDER'S DECOMPOSITION

We frame our solution around a classical optimization concept known as the "(generalized) Bender's decomposition" (Geoffrion, 1972). Consider the following optimization problem

$$\min_{x,y} f(x,y) \tag{3a}$$

$$\text{s.t. } g(x,y) \le 0 \tag{3b}$$

$$x \in X, y \in Y \tag{3c}$$

where we call $y$ *complicating variables*. A complicating variable is a variable that, if fixed, makes the rest of the optimization much easier. For instance, the problem $\min_{x,y}(\sin(y) - x)^2$ becomes trivial if we first fix $y$ to any value. We define the constrained set $\mathcal{C} = \{(x,y) \in X \times Y | g(x,y) \le 0\}$.

Bender's decomposition splits this optimization problem into a *master problem* and a *subproblem*. The master problem proposes solutions to the problem in $y$, ignoring the impact of the choice of $x$. The subproblem then uses the solution $y$ from the master to solve for the remaining variables $x$.

---

[1]In fact, it might be the case that such a tour does not exist which is an undetectable case for Bello et al. (2016). Generally, deciding whether such a tour exists is already NP-complete (Held & Karp, 1965)

Figure 2: Sketch of the policy evaluation. First, a classical solver (like SCIP (Bestuzheva et al., 2023) is used to solve the current optimization problem composed of the static $\mathcal{C}$ constraints (blue) and the dynamically added inequality constraints (purple). The current solution (red) is first passed to the environment to produce the reward, and additional features $f_i$. The solution produced by the solver, combined with the constraints $\mathcal{C}$ and additional features $f_i$ are passed to the policy-GNN $\pi$, which predicts a new inequality constraint (red). This inequality constraint is overlaid onto the existing optimization problem and the classical solver finds the next solution candidate $x_{i+1}^\star$.

Based on the value and feasibility of the subproblem we then add additional linear constraints (so called "cuts") into the master problem and repeat the optimization with the additional constraint.

Specifically, we can distinguish *feasibility cuts*, which remove items from the master problem that do not lead to a solvable subproblem, and *optimizing cuts* which manipulate the objective function of the master problem to steer it towards better solutions. To control the objective function, one classically adds an auxilliary variable $\varphi$ that is lower bounded by cutting information from the subproblem. Schematically, the master problem looks like

$$\min_y f(y) + \varphi \tag{4a}$$

$$s.t.\ g(y) \leq 0 \tag{4b}$$

$$\varphi \in \mathcal{O}(x,y), y \in \mathcal{F}(x), x \in X, y \in Y, \tag{4c}$$

where $f(y)$ and $g(y)$ are lower bounds in $x$ to $g(x,y)$ and $f(x,y)$ respectively, and $\mathcal{O}, \mathcal{F}$ are additional constraints that are generated by solving a subproblem (see Geoffrion (1972)). For the sake of this work, we will only consider optimality constraints $\mathcal{O}(x,y)$.

## 4   BENDER'S ORACLE OPTIMIZATION

Ordinary optimality cuts have the form of

$$\varphi \geq z(x^*) + \lambda^T \nabla_x g(x^*, y^*)(x - x^*), \tag{5}$$

where $z(\cdot)$ is the result of the subproblem $z(x^*) = \min\{c^T y : g(x,y) \leq 0, y \geq 0\}$ conditioned on the solution $x^*$ of the master problem, $\lambda$ is the optimal dual solution, $y^*$ is the solution to the subproblem, and $\varphi$ is a helper variable that is added to the objective $\max c^T x + \varphi$. This has a nice interpretation of placing a lower bound on the main problem based on the linearization of the subproblem around the current optimum (see, e.g. Geoffrion (1972)).

Our core insight is the following: Notice that we do not necessarily need to solve the subproblem: We only need to know what constraints (Eq. 5) the subproblem *would* add, if we solved it to completion, but we do not need to actually solve the intractable subproblem. Therefore, if we had a function that takes a feasible point and generates the cutting plane coefficients similar to Eq. 5, we could optimize the original problem, while always staying safe due to the master problem handling the constraints. Bender's Oracle Optimization (BOO) directly learns a scalar corresponding to the bias $b = z(x^*) + \lambda^T \nabla_x g(x^*, y*)x^* \in \mathbb{R}$, and a vector corresponding to the linear weight $w = \lambda^T \nabla_x g(x^*, y*) \in \mathbb{R}^d$ without explicitly constructing and solving the underlying optimization problem(s). We learn both of these values using RL with feedback from a simulator.

This gives us the following MDP: States are given by the current solution $x_i^\star$ and static features (e.g. for a scheduling problem "what jobs exist for scheduling"). Actions are the parameters $(w, b)$ which parameterize the linear inequality. The rewards and transition function is given by the black box environment. Notice that this parametrization allows us to solve a CMDP with known constraints using ordinary MDP algorithms, while maintaining guaranteed safety during training and inference.

BOO recursively refines its answer $x^\star$ by adding bender's cuts to the problem, depending on the previous iteration's result $x_k^\star$ and finds the optimal $x^\star$ from the set of possible results. We define BOO recursively as

$$x_0^\star = \operatorname{argmax} \mathcal{S}_0 = \operatorname{argmax}\{\varphi | x \in \mathcal{C}\} \tag{6a}$$

$$x_1^\star = \operatorname{argmax} \mathcal{S}_1 = \operatorname{argmax} \mathcal{S}_0 \cap \{\varphi | \varphi \geq b_1 + w_1^T x, \ (b_1, w_1) \sim \pi(b, w | x_0^\star, \mathcal{C})\} \tag{6b}$$

$$x_2^\star = \operatorname{argmax} \mathcal{S}_2 = \operatorname{argmax} \mathcal{S}_1 \cap \{\varphi | \varphi \geq b_2 + w_2^T x, \ (b_2, w_2) \sim \pi(b, w | x_1^\star, \mathcal{C})\} \tag{6c}$$

$$\cdots$$

$$x_k^\star = \operatorname{argmax} \mathcal{S}_k = \operatorname{argmax} \mathcal{S}_{k-1} \cap \{\varphi | \varphi \geq b_k + w_k^T x, \ (b_k, w_k) \sim \pi(b, w | x_{k-1}^\star, \mathcal{C})\}. \tag{6d}$$

The variable $\varphi$ acts as the quality estimate for the result $x$. We can see the core assumption of BOO in Eq. 6: We assume that we have an optimizer that can find a feasible point within $\mathcal{C}$. BOO refines the estimate of the optimal decision $x^\star$ online by recursively improving the value estimate using cuts sampled from the BOO policy $\pi$. If a cut at step $i > 0$ produces an unsatisfiable problem, we backtrack and sample a new inequality.[2] Since only linear constraints are added by BOO, the subproblems in each iteration can usually be solved very quickly using off-the-shelf solvers such as SCIP (Bestuzheva et al., 2021) or IPOPT (Wächter & Biegler, 2005). However, even if no explicit mathematical expression for $\mathcal{C}$ is present we can still apply BOO since we only need to be able to sample feasible points $(x, \varphi)$. Formally, we can prove

**Theorem 1.** *Given a feasible set $\mathcal{C}$, the recursive BOO scheme Equation* (6) *will either*

- *Produce a sequence $x_i^\star \in \mathcal{S}_i$ such that all $x_i \in \mathcal{C}$*

- *return "unsatisfiable" if* **C** *is empty.*

The full proof can be found in Appendix D. This allows us to conclude: BOO is always safe.

Intuitively, our method creates a new unconstrained MDP across the number of cutting planes $k$ rather than solving a CMDP across the time dimension. We parameterize $\pi$ as a Graph Neural Network (GNN) (Kipf & Welling, 2016) connecting *variable nodes* with *constraint nodes* as is done in prior work (see e.g, (Labassi et al., 2022))

---

**Algorithm 1** Bender's Oracle Optimization.

**Input:** training constraint sets $C = \{\mathcal{C}_0, \ldots, C_N\}$, classical optimizer $OPT$, environment to evaluate feasible points, number of cuts $K$
**repeat**
    sample instance and constraints $\mathcal{C}_i \sim C$
    initialize $\mathcal{S}_0 \leftarrow \mathcal{C}_i$ and $f_0 \leftarrow \emptyset$
    Use $OPT$ to find initial feasible point $x_0^\star \in \mathcal{S}_0$
    if infeasible, return "unsatisfiable"
    **for** $k = 1$ **to** $K$ **do**
        predict $\pi(b, w | x_{k-1}, \mathcal{C}_i, f_i)$
        build $\mathcal{S}_k$ using $(b, w)$ and $\mathcal{S}_{k-1}$ (Eq. 6)
        solve $x_k^\star \in \mathcal{S}_k$ using $OPT$
        evaluate $x_k^\star$ in environment to get features $f_{i+1}$ and reward $r_k$ for $x_k^\star$
        add stochastic constraints to $\mathcal{S}_k$
        record $(b_k, w_k, x_k^\star, r_k, f_{i+1})$
    **end for**
    update $\pi$ on $x_k^\star, r_k, (b_k, w_k), f_{i+1}$ using PPO
**until** convergence

---

[2]This happens rarely during the first training iterations where cuts are random.

We provide the training algorithm in Algorithm 1. First, a random instance $\mathcal{C}_i$ is sampled from a set of instances (or a random generator). We directly solve the feasibility problem $x_0^\star \in \mathcal{C}_i$, which gives us our starting point $x_0^\star$. If no point within $\mathcal{C}_i$ exists, we can immediately determine that no optimal solution exists. If a point is found, $x_0^\star$ is - by definition - feasible, but usually not optimal. The RL algorithm then repeatedly refines the estimate for the solution $x_i^\star$ by looking at the current solution $x_{i-1}^\star$ and producing new inequalities that "cut" of the bad solutions according to Equation (6). Every time a new solution is produced, it is checked against the environment, which returns rewards and features for the next cutting step. During training, we record all added inequalities (actions) and solutions $x_i^\star$ and features $f_i$ (states), which we are used for training using PPO (Schulman et al., 2017). Inference uses the same scheme updating $\pi$.

## 5 EXPERIMENTS

To our knowledge there are three ways of solving general design problems: Sequential bit-placement methods using pointer networks (Bello et al., 2016), classical optimization (assuming no stochastic effects and white-box access to the model), and our method. We use a standard graph convolutional neural network (GCNN) (Kipf & Welling, 2016) with global and edge features, a hidden dimension of 128, and 4 message passing steps. For optimization we use the Adam optimizer and standard PPO hyperparameters.

We compare these solvers on two problem sets. First, we study learning a nonlinear and nonconvex objective function over a nontrivial feasible set, but without considering stochasticity. Since we can set this problem, we can compare against the global optimum as found by the SCIP global nonlinear optimizer (Bestuzheva et al., 2023). We also demonstrate the performance advantage over solving the original MINLP: Our solver is capable of learning a parametrization of the problem that is nearly $800\times$ faster to solve than the true parametrization.

Second, we consider a large-scale stochastic job scheduling problem, where the objective is to maximize the profit of a set of jobs, each consisting of a set of operations that have to be completed in order, within a limited time. A job only receives profit if all its operations are completed in the correct order by the time of completion. We add stochasticity to the problem, by having a set of task-types that determine how likely a job is delayed and how much profit is to be made by completing the job. This means our agent has to learn a complex risk-reward tradeoff, while also having to produce feasible job scheduling plans.

### 5.1 NONCONVEX CONSTRAINED PROBLEM

To estimate the ability of our method to recover a nonlinear objective function over a constrained set, we consider the following toy problem

$$\max_x x^T A x + b^T x + c \tag{7a}$$

$$k^T x \le p \tag{7b}$$

$$x \in \{0, 1\} \tag{7c}$$

where $A$ is a random positive semidefinite matrix, $b$ and $k$ are random vectors, and $k$ is a random constant and $c$ is an offset always set to $c = 1$. This type of problem is frequently found in economics where many problems can be reduced to convex maximization over binary variables subject to linear constraints (see Zwart (1974)). There are also applications to machine learning like, for instance, non-negative sparse PCAs (Zass & Shashua, 2006) or feature selection (Mangasarian, 1996).

We use this model as input to the global optimizer in SCIP (Bestuzheva et al., 2023), but *hide* the objective for our RL agent. The goal of our agent is to find optimizing cuts, such that the found $x$ maximizes the hidden $x^T A x + b^T x + c$ while staying feasible.[3] We give all methods two input features: First, we give the diagonal value of $a_{i,i} \in A$ for every variable $x$. Second we give the row/column sum $\sum_{i=0}^N a_{i,j}$ for every variable. Generally, this is *insufficient to reconstruct the entire objective* function. This means the problem is a Constrained Partially Observable Markov Decision Process (CPOMDP), where the model has to gather information from the found solutions $x_k$.

---

[3] Notice that this problem is *not convex* since we maximize over a convex function rather than minimize (see, e.g, Zwart (1974)).

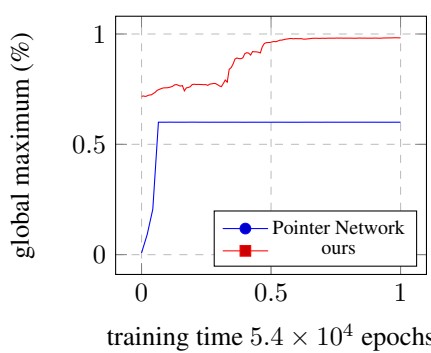 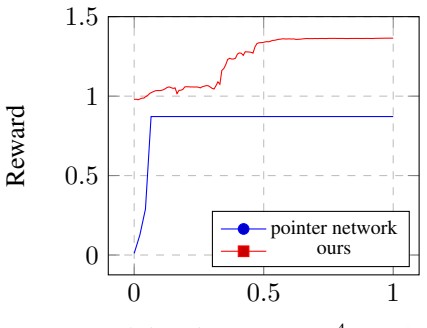

Figure 3: Validation performance of a pointer network compared to BOO on our nonconvex combinatorial problem. The x-axis is normalized towards the number of problems as the methods use different numbers of steps per problem instance (our method uses dramatically fewer steps).

Table 1: Comparisons of the pointer network and our method on unknown instances of the nonconvex optimization problem with one or two inequality constraints. We showcase the quality of the found solution as a percentage of the global optimum, and the time needed to find that solution. For the setting with 2 inequality constraints, we truncate the MINLP solver after 120s.

|  |  | % global maximum | time policy | expected regret |
|---|---|---|---|---|
| | **ours** | **0.98** | **0.07**s | **0.11** |
| 1 constraint | pointer network | 0.60 | 1.10s | 0.43 |
| | global MINLP | 1.0 | 60.19s | 0.0 |
| | **ours** | **0.95** | 7.98s | **0.06** |
| 2 constraints | pointer network | 0.45 | **0.55**s | 0.55 |
| | global MINLP | 1.0 | >120s | 0.0 |

As a baseline, we utilize a variant of the pointer network (Vinyals et al., 2015) used in Bello et al. (2016) with the difference that instead of a simple unstructured RNN (Schmidt, 2019), we use exactly the same GNN backbone as in our method to make sure no method is disadvantaged by a smaller/larger network or different data availability.

We further compare against a naive baseline where we optimize Eq. 7a by linearizing the objective around 0. Notice that both this linearized model and our BOO model can be efficiently optimized with LP-solvers, while the original objective has to use much more complex MINLP solvers. As our reward we compare the quality of the solution found by our policy against its linearization:

$$R = \frac{x_\pi^T A x_\pi + b^T x_\pi + c}{x_b^T A x_b + b^T x_b + c}, \tag{8}$$

where $x_\pi$ is the solution found by BOO, and $x_b$ is the result found by maximizing the linearized objective. $R > 1$ means our model exceeds the naive baseline, while $R < 1$ implies the model is worse than the linearized objective.

As we can see in Fig. 3, our method manages to reach almost the $100\%$ of true objectives value after roughly half the exploration budget has been reached. The pointer network quickly reaches a saturation level of roughly $60\%$ of the global maximum.

Looking into Table 1, we can also see that our method tends to find solutions orders of magnitude faster than the MINLP solver that knows the objective function with minimal loss in quality. This is not entirely unsurprising as Bender's decomposition is fundamentally a way of speeding up MINLP problems (see Section 3 or Geoffrion (1972)), but it is nevertheless interesting to see that this property translates to black-box learning of objective functions. We also noticed that as our method improves, it tends to learn policies that find optima faster (see Appendix B).

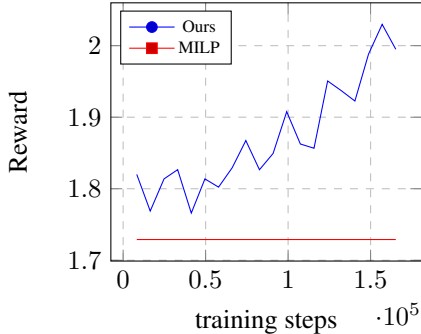

Figure 4: Performance of our model on the stochastic Scheduling problem compared to optimal solutions provided by a MILP solver. The performance is shown over a *unseen* validation set

We also study the same problem with two constraints. This problem is significantly more difficult, so much so that finding the global optimum is generally infeasible. Therefore we allow SCIP a time budget of 120s and compare the found solution against ours and the pointer network. The results can be found in Table 1. Notice that our method performs similarly to the single constraint version, still achieving near optimal performance, while the pointer network drops under 50% of the global performance. Further, notice that the time taken by our policy increases as the problem difficulty increases, since the new optimization problem is significantly harder than the one with a single constraint. This stands in contrast to the pointer-network driven solution, which allocates the same amount of compute to every decision, regardless of how complex the constraints are.

One advantage of a properly constrained RL agent is that one can train during deployment without having catastrophic failures in safety. Therefore we also report the regret (i.e., the area under the performance curve in Fig. 3) one would expect when training this agent online in Table 1. As one can see our agent outperforms the pointer network by close to $4\times$.

## 5.2 SCHEDULING PROBLEM

We consider the problem of finding a schedule that maximizes returns within a fixed timeframe. In our setup, we consider 3 different machine types, where each machine has $M$ duplicates. We sample $J$ jobs that have a randomly sampled expected completion time for each machine. The machines have to be worked on in order: First machine 1, second machine 2, third machine 3. We have a time-limit of $T$ days to finish as many jobs as possible. A job only pays out its profit if all of its operations were completed in time and in the right order. Informally, we can model a schedule as $M \times J \times T$ binary variables where 1 indicates that Job $j$ is scheduled on machine $m$ at timestep $t$. This corresponds to a variation of the time-indexed scheduling problem seen in e.g. Ku & Beck (2016). The constraints ensure that a job is worked in the correct order, no machines are double-booked, etc. The resulting constraint set is orders of magnitude more complex than what is usually consider in black-box CMDP solvers and, just like the previous nonconvex toy problem, not solveable with off-the-shelf CMDP solvers.[4] We give the full constraint set in Appendix C.

Within this feasible set, every assignment of $x_{m,j,t}$ corresponds to a plan that is expected to be feasible. To simulate random events, we unroll the plan recursively and simulate delays by randomly extending the time taken for a scheduled operation $o(j,m)$ by between 1 and 3 months. The probability for a delay depends on the job class $C(j)$, which is randomly assigned to jobs as a feature. After a random event, the schedule is invalid and has to be re-generated from that day forward.

The profits are similarly hidden, but also depend on the job class $C(j)$, such that a riskier job obtains a higher payoff. Since the likelihood of a job being delayed is predictable, the model should be able to learn a robust plan that outperforms the optimal schedule where jobs are planned independent of their probability of success or the expected reward for that operation. This gives a highly complex risk-reward tradeoff where one has to balance risky but high profit jobs against lower risk, but lower profit jobs. For our experiments we choose $T = 12$, $J = 200$ and $M = 4$.

---

[4]We were not able to get CPO to converge to within the constraint set.

Table 2: Comparison of our method against an optimal MILP solver (higher is better).

|      | Reward@$0.5 \times 10^5$ | Reward@$1.0 \times 10^5$ | Reward@$1.5 \times 10^5$ |
| --- | --- | --- | --- |
| Ours | **1.80** | **1.87** | **2.03** |
| MILP | 1.73 | 1.73 | 1.73 |

Since, to our knowledge, no solver for this stochastic planning problem exists, we follow the approach taken by (Fawzi et al., 2022; Mankowitz et al., 2023) and compare ourselves against a classical MIP formulation that plans optimally with the information it has, and re-plans in the case of a stochastic event. In our case, this involves solving the underlying mixed-integer linear program *without* considering stochastic effects and knowledge of the true value of each job. As a reference value, we solve this model as a baseline to $\max \sum_{j=1}^{J} y_j$, which can be seen as an uninformative prior, where all stochastic and (nonlinear) profit functions are ignored, in favor of simply packing the schedule as tightly as possible. We do not use the true (hidden) job-rewards as the objective function since that would cause the MILP to plan all high reward, but also high-risk jobs (which is highly suboptimal). For our agent to beat the baseline, it has to be able to deal with stochastic effects, and has to learn the true value of completing a job. For this, we set up the job values as the likelihood of a job being interrupted, i.e., if a job has probability $0.9$ of being delayed at any specific point in time, the reward for completing it is $0.9$. This gives a natural risk-reward structure, where riskier jobs yield more reward.

The results for this can be found in Fig. 4 and Table 2. As we can see our method quickly exceeds the performance of the greedy MILP solver. Since our method always returns a valid schedule, this method can be used as a drop-in replacement for traditional MILP solvers when feedback from the environment is available. Since our method can be trained during deployment, it makes sense to also consider the advantage of our method against the baseline. Our method offers an expected improvement over the training interval (Fig. 4) of $\frac{\int \text{ours}(t)dt}{\int \text{base}(t)dt} \approx 8.2\%$. Note that this depends on the training time since longer training mean the model spends more time in the RL-optimized region.

## 6 LIMITATIONS

Our method relies on an existing classical optimizer to allow for training with highly complex constraint sets. This has the advantage of inheriting the work that has gone into building state-of-the-art optimization algorithms. However, it also inherits many of the challenges from classical optimization, such as the necessity of formulating all problems in an algebraic modeling language. The method proposed here is mostly targeted to applications where one has access to the constraints. In cases where such a model is not available, one would need to train such a model during learning, turning this algorithm into a model-based RL approach. We leave the combination of model learning and BOO for future work.

## 7 CONCLUSION

We propose a general method for converting challenging design problems into a Constrained Reinforcement Learning problem with known constraints. To solve the resulting problem we introduce a new algorithm "BOO" that finds optimal points within a feasible set by iteratively adding linear constraints to locate the optimal solution of the problem. Since our policy only adds linear inequality constraints our method remains solvable using high performance classical optimizers. Our method also guarantees that constraints are strictly satisfied during training, allowing us to train our method online even in safety critical scenarios.

We showcase the abilities of our method in a synthetic combinatorial environment, and a job-scheduling problem. Our method shows superior performance over both a MILP and neural-network baseline, while offering drastically faster convergence compared to a MINLP solver, in cases where an analytical expression exists. To our knowledge this is the first reinforcement learning method that allows arbitrary constraints to be enforced during training and inference.

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

# A LLM USAGE

We used LLMs to improve the writing and presentation of the work, as well as to aid in translation. We manually ensured that any alteration in wording did not change the meaning of the work.

# B OPTIMIZATION SPEED OVER TIME

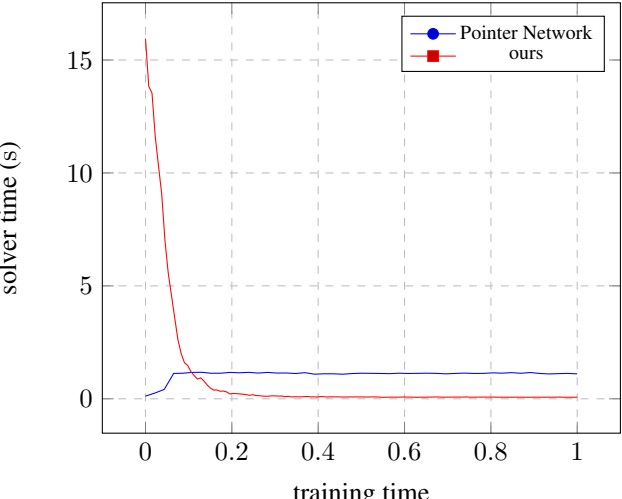

Figure 5: Time taken to find a solution.

We find that BOO implicitly learns to find solutions more efficiently. We assume this is because we impose a 60s time budget on finding solutions during training time, since this is the expected solving time for our problem class. This might implicitly regularize found policies towards simpler solutions as more complex solutions run the risk of not being solvable to global optimality within the time budget. We assume this effect could be increased by explicitly including training time in the objective, but investigating this is left for further research.

The reason that the pointer network increases in time to find a solution is because the model learns to take advantage of the existing budget given by $k^T x \leq b$, which means it can place more items $x_i = 1$ into the feasible set, which then implies that one has to roll out the RNN over more timesteps, leading to slower inference.

# C FEASIBLE SCHEDULES

The set of feasible schedules is given by: Let $y_j$ be a binary indicator of whether job $j = 1, \ldots, J$ is worked to completion, $x_{m,j,t}$ be the binary indicator of whether job $j$ is scheduled on machine

$m = 1, \ldots, M$ at timestep $t = 1, \ldots, T$.

$$\sum_{t=1}^{T} x_{m,j,t} \leq 1 \qquad \forall m = 1 \ldots M \forall j = 1 \ldots J \qquad (9a)$$

$$y_j \leq \sum \sum_{t=1}^{T} x_{m,j,t} \qquad \forall m = 1 \ldots M \forall j = 1 \ldots J \qquad (9b)$$

$$\sum_{t=0}^{T} (t + \text{jobtime}(j)) x_{m,j,t} \leq T \qquad \forall m = 1 \ldots M \forall j = 1 \ldots J \qquad (9c)$$

$$\sum_{j=1}^{J} \sum_{t'=t-\text{jobtime}(j)+1}^{t+1} x_{m,j,t'} \leq M \qquad \forall m = 1 \ldots M \forall t = 1 \ldots T \qquad (9d)$$

$$\sum_{t=0}^{T} (t + o(j, m-1)) x_{m-1,j,t} \leq \sum_{t=0}^{T} t x_{m,j,t} \qquad \forall m = 2 \ldots M \forall j = 1 \ldots J \qquad (9e)$$

$$\sum_{t=0}^{T} x_{m-1,j,t} \geq \sum_{t=0}^{T} x_{m,j,t} \qquad \forall m = 2 \ldots M \forall j = 1 \ldots J \qquad (9f)$$

$$\text{jobtime}(j) = \sum_{m=1}^{M} o(j, m) \qquad (9g)$$

$$y_j, x_{m,j,t} \in \{0, 1\} \qquad (9h)$$

where $T$ is the global timelimit, and $o(j, m)$ is the time job $j$ takes on machine $m$. Equation (9a) makes sure every operation is only scheduled once, Equation (9b) sets the auxiliary variable $y_j$ denoting whether a job $j$ is completed in time, Equation (9c) makes sure that all scheduled operations complete within the timelimit, Equation (9d) prevents two operations being scheduled on the same machine simultaneously, Equation (9e) makes sure that operation $m$ of job $j$ happens after operation $m - 1$, and Equation (9f) makes sure that if machine $m$ is scheduled, machine $m - 1$ also has to be scheduled. This is a highly constrained MILP problem, meaning that randomly generating a plan $x_{m,j,t}$ is almost always going to be infeasible according to the constraints Equation (9).

## D  SAFETY PROOF

*proof of theorem 1.*  BOO paramterizes the solution $x_i^\star$ as the recursive intersection of sets. We will prove the stronger statement that all $x \in \mathcal{S}_i$ (not only the optimal one) are safe for all $i = 0, \ldots, \infty$

Proof via induction:
Base case: For $x \in \mathcal{S}_0$, we have $x \in \mathcal{C}$ by definition, unless $\mathcal{C} = \emptyset$ in which case we can immediately return "unsatisfiable".
Induction hypothesis: $x \in \mathcal{S}_k \implies x \in \mathcal{C}$.
Induction step: For every subsequent iteration, we have $\mathcal{S}_{k+1} = \mathcal{S}_k \cap \{\varphi | \varphi \geq b_k + w_k^T x\}$ therefore $\mathcal{S}_{k+1} \subseteq \mathcal{S}_k$. Meaning that
$$x \in \mathcal{S}_{k+1} \implies x \in \mathcal{S}_k \implies x \in \mathcal{C}.$$

We conclude all $x \in \mathcal{S}_i$ $x$ is safe, therefore $x_i^\star$ is safe. $\qquad \square$

