# OpenReview forum: "Constrained Reinforcement Learning using Bender’s Decomposition and Exact Constraint Satisfaction"
_ICLR.cc/2026/Conference — ICLR 2026 Conference Withdrawn Submission_

### Official Review · Reviewer_JvvW · 2025-10-29

**Soundness:** 1
**Presentation:** 1
**Contribution:** 3
**Rating:** 2
**Confidence:** 4

**Summary:**

The paper proposes a method based on Bender's decomposition to solve constrained combinatorial optimization problems. As far as I can tell, given an initial feasible solution (which must be generated by an existing method), feasibility cuts are generated by something, I don't really know what, but whatever it is it is learned through RL. The approach (BOO) is tested on a non-linear optimization problem (there are literally no more details than this, other than that the A matrix is semidefinite) and some kind of scheduling problem (again, I cannot describe it because the authors do not even provide a name from the literature). A comparison is made versus the pointer network from Bello et al. (2016).

To summarize my views on this paper pre-response: the idea sounds great, the problem is important, but I have no idea what is going on in most of the rest of this paper.

**Strengths:**

1. The setting is one of great importance: we need RL methods that respect constraints to solve highly constrained MI(N)LPs. The literature currently offers essentially nothing that can solve these problems other than playing with penalization of violated constraints, which does not work well.
2. The method proposed is theoretically sound, assuming the (w, b) pairs generated are valid. I suppose this is the case, but I really do not understand from this paper why it is the case.
3. I find it hard to say much about the performance, but at least on the optimization problem on "unknown instances" the pointer network is outperformed.

**Weaknesses:**

1. The primary issue with this paper is that I do not understand the method. I am well-versed in Benders decomposition and in reinforcement learning. Algorithm 1 is rather high level (with some minor issues, see below), but the main issue is this: where does (b, w) come from and why can we guarantee that this cut is valid? The point that we do not need to solve the Benders subproblem to optimality is well-taken, and I note, well-known in the OR literature. I am missing, however, is what then happens to generate this point. I hope the authors can somehow clear this up in the response.
2. The experimental evaluation in this paper is rather unique, and not in a good way. The "nonconvex constrained problem" has practically no details provided, and what even are the constraints? The type of constraints matter, perhaps even more than the number of constraints. This whole section is so vague that I do not really know how to interpret it. I do not even know how the pointer network can be used to solve the problem given. But this leads to the next point...
3. The pointer network used is very old. It is indeed the basis for future work, but beating the pointer network from Bello et al. is really not an accomplishment worthy of ICLR.
4. The scheduling problem provided feels almost randomly pulled from the literature. Why should we care about this random problem? The formulation used is not even a good one -- the paper the authors cite (Ku & Beck) literally says in the abstract that there are better formulations. Then there is a dynamic/stochastic aspect that is added, but it is not clear to me exactly how it even works. Furthermore, I do not understand the experimental results on this problem (Table 2) at all. There is a wide literature on neural combinatorial optimization, please see any of these papers to see how to compare two algorithms to one another. I also note that further baselines are necessary.

Minor notes:
1. In the introduction, the authors write about infeasible dead-ends. The example given regarding a sparsely connected graph is not correct. For a TSP with no additional side constraints, a sparse graph will not lead to infeasibility unless the problem itself is infeasible. If we had, e.g., time windows, then infeasibility is possible. The principle is of course correct, the example just needs tweaking.
2. On page 4, some of the y*'s have the asterisk not in the superscript.
3. Algorithm 1 line "add stochastic constraints" this is the first time the paper talks about stochastic constraints.
4. The paper uses quite a bit of space to vaguely describe the scheduling problem (page 6 and page 8) and might as well show the model rather than beating around the bush.
5. Regarding the model in Appendix C, it should say "Constraints" or "Inequalities" not "Equality"
6. Table 2: which MILP solver is being used here?
7. The discussion on the top of page 9 about the stochastic planning problem: indeed no "solver" for this problem exists since it seems to be completely made up rather than taken from the literature, for which solvers will exist.

**Questions:**

1. See question above regarding the generation of (b,w)
2. Why is it interesting that you beat this old pointer network?
3. Where are any other interesting baselines?

---

### Official Review · Reviewer_xngN · 2025-11-01

**Soundness:** 2
**Presentation:** 2
**Contribution:** 2
**Rating:** 2
**Confidence:** 3

**Summary:**

Reinforcement Learning (RL) has been recently applied to non-sequential decision making tasks including matrix decompositions, algorithm discovery, and combinatorial optimization through the Constrained Reinforcement Learning (CRL) framework. Traditionally, CRLs are modeled using Constrained Markov Decision Process (CMDPs). However, CMDPs does not allow for hard constraints, as it enforces the constraints in expectation over trajectories instead of per trajectory.

The paper proposes to use the Bender’s Decomposition framework, which divides a problem into a master problem and a subproblem based on complicating variables. The subproblem can be used to propose optimality cuts, which identifies better solutions without affecting their feasibility. The paper then introduces Bender’s Oracle Optimization (BOO), which learns to generate Bender’s optimality cuts using RL without affecting feasibility. Experiments show that BOO achieves near optimal results in significantly shorter time than the classical solver SCIP.

**Strengths:**

Bender’s decomposition is a well established technique for iteratively solving a difficult integer programming problem, it is interesting to see how it can be used in combination with a machine learning approach.

The experiments show promising results as the proposed method converges much faster than the classical solver SCIP.

**Weaknesses:**

The paper is difficult to follow, specifically, the connection from Constrained RL to Benders Decomposition is not very clear. AlphaTensor is used to motivate the method a lot in the introduction (line 92 states that the method can be used to reframe alphatensor). But the rest of the paper makes no reference to it. It would be useful to see how a reformulation can take place to make the connection clearer.

The paper refers to “design problems” without a clear definition. Specifically, the experimental results only compare to pointer networks and classical solvers, and claim that these are the only 2 ways of solving general design problems. A clearer definition would allow better support for the claim. It is also unclear how the nonconvex constrained problem and the scheduling problem represents design problems.

The paper describes the guaranteed satisfiability at every state as a major advantage of the framework. However, this guarantee comes from the fact that the state at each step comes from an exact classical optimizer since the actions of the proposed framework simply add optimality cuts to the master problem. Wouldn’t adding an exact classical optimizer and the exact feasibility constraints to other CMDP approaches also enable guaranteed satisfiability?

**Questions:**

The subproblem fixes the complicating variables in Benders Decomposition, and becomes much simpler to solve, as the paper points out. What are then the benefits to bypassing the solving step and approximating the cuts?

Did the author test with classical solvers other than SCIP?

Benders Decomposition utilizes the Dual of the Subproblem to add cuts while BOO learns to generate the cuts directly. Can the framework also utilize the dual?

Misc:

Line 214 equation has a small formatting error.

Table 1 should specify what the bolded values are.

---

### Official Review · Reviewer_Jt8Y · 2025-11-01

**Soundness:** 3
**Presentation:** 3
**Contribution:** 3
**Rating:** 4
**Confidence:** 3

**Summary:**

The paper introduces Bender’s Oracle Optimization (BOO), a framework for constrained reinforcement learning with hard, per-instance feasibility guarantees. Instead of acting in a sequential action space that can reach dead ends, a GNN policy proposes linear cutting planes (w,b) that are added to a master optimization problem; a classical solver (e.g., SCIP/IPOPT) then re-solves while always remaining within the known constraint set C. This couples RL with Bender’s decomposition: the policy learns optimality cuts without solving the subproblem, while the solver enforces feasibility. A safety theorem (Theorem 1) shows that BOO never leaves C is nonempty. Two examples are shown in the experiments.

**Strengths:**

1. Hard-constraint satisfaction is guaranteed by construction via the master problem and linear cuts (Theorem 1).
1. Recasts non-sequential design/optimization tasks as constrained RL over known C, sidestepping CMDP “in-expectation” limitations
1. Leverages mature MIP/MINLP solvers for feasibility while learning problem-specific optimality guidance
1. Strong gains vs. pointer networks on nonconvex combinatorics and vs. MILP baseline under stochasticity; notable speedups vs. direct MINLP

**Weaknesses:**

1. BOO adds only linear inequality cuts; highly nonlinear subproblem structure may require many cuts or limit convergence quality. Unfortunately, this is not explored or tested via experiments.
1. It relies on outside solvers. Practical performance hinges on solver quality and modeling effort; repeated solves per cut can be computationally heavy at larger scales.
1. The paper does include meaningful baselines (pointer network, MINLP/MILP), but the heuristic coverage is limited—useful for a first look, yet not exhaustive for the target domains.
1. Experiments are on one synthetic nonconvex problem and one scheduling setup; broader benchmarks (incl. stronger CRL baselines beyond pointer networks/MILP) and ablations (e.g., number of cuts K, policy features) would strengthen claims.

**Questions:**

1. What's the training cost for the RL model (hours or hardware)?

---

### Official Review · Reviewer_PiVQ · 2025-11-04

**Soundness:** 4
**Presentation:** 3
**Contribution:** 3
**Rating:** 8
**Confidence:** 2

**Summary:**

This paper introduces Bender's Oracle Optimization (BOO), a framework that reformulates non-sequential optimization tasks as constrained reinforcement learning problems by learning to generate cutting planes that refine the solution space. The method ensures constraint satisfaction throughout training by delegating constraint handling to classical optimizers like SCIP, enabling safe learning during deployment. BOO demonstrates effectiveness on two complex problems: a nonlinear, nonconvex packing problem achieving near-globally optimal solutions up to 800× faster than traditional solvers, and a stochastic job-shop scheduling problem where it outperforms mixed-integer linear programming by learning to account for stochastic effects. The framework represents the first reinforcement learning approach that enforces arbitrary constraints during both training and inference while maintaining strict feasibility guarantees.

**Strengths:**

1. The paper addresses a hard problem -- maintaining constraints in reinforcement learning both during training and inference
2. Using Bender's decomposition is a novel idea that has not been explored before
3. Clear evaluations against prior solutions and classical solutions in MILP. Proposed solution is faster and gets higher performance compared to state-of-the-art.
4. Theoretical guarantee for safety

**Weaknesses:**

1. The language used can be simplified to be accessible to a wider audience
2. Constraints are known beforehand, not true in realistic settings
3. No convergence or optimality guarantee
4. Learned policy is not explainable

**Questions:**

- I wonder what happens if you use approximate constraints, would the solution still converge to optimal?
- Given that added constraints change the optimization landscape drastically, can you guarantee reaching the optimal solution theoretically? Is it possible that the policy never converges?
- How do you determine the number of cuts? Is the solution sensitive to the choice?
- How does your work differ from these works?
[1] Huang, Zeren, et al. "Learning to select cuts for efficient mixed-integer programming." Pattern Recognition 123 (2022): 108353.
[2] Paulus, Max B., et al. "Learning to cut by looking ahead: Cutting plane selection via imitation learning." International conference on machine learning. PMLR, 2022

---

### Note · Authors · 2025-11-25

**Comment:**

Thank you very much for the valuable feedback provided by the reviewers. We greatly appreciate the time and effort invested in evaluating our submission.
We realise that this paper needs a more changes than are possible in the narrow reviewing window and we have therefore decided to withdraw our paper from ICLR.
We nevertheless want to thank the reviewers for their time reviewing this work which will help us revise our work.

Thank you again for your consideration and support.

**Withdrawal Confirmation:**

I have read and agree with the venue's withdrawal policy on behalf of myself and my co-authors.